# Diagnostic Accuracy of Touchscreen-Based Tests for Mild Cognitive Disorders: A Systematic Review and Meta-Analysis

**DOI:** 10.3390/diagnostics15182383

**Published:** 2025-09-18

**Authors:** Nathavy Um Din, Florian Maronnat, Bruno Oquendo, Sylvie Pariel, Carmelo Lafuente-Lafuente, Fadi Badra, Joël Belmin

**Affiliations:** 1Laboratoire d’Informatique Médicale et d’Ingénierie des Connaissances en e-Santé (LIMICS), Université Sorbonne Paris-Nord, 75005 Paris, France; nathavy.umdin@gmail.com (N.U.D.); badra@sorbonne-paris-nord.fr (F.B.); 2Centre Hospitalo-Universitaire de Brest, Université de Bretagne Occidentale, 29200 Brest, France; florian.maronnat@chu-brest.fr; 3Hôpital Charles Foix, 7 avenue de la République, 94200 Ivry sur Seine, France; bruno.oquendo@aphp.fr (B.O.); sylvie.pariel@aphp.fr (S.P.); carmelo.lafuente@aphp.fr (C.L.-L.); 4Faculté de Santé, Sorbonne Université, 91-105 Boulevard de l’Hôpital, 75013 Paris, France; 5Clinical Epidemiology and Ageing (CEpiA) Team, Université Paris Est-Créteil, INSERM, IRMB, 94010 Créteil, France

**Keywords:** older adults, mild neurocognitive disorder, mild cognitive disorder, touchscreen, diagnosis, digital tools

## Abstract

**Background/Objectives**: Mild neurocognitive disorder (mNCD) is a state of vulnerability, in which individuals exhibit cognitive deficits identified by cognitive testing, which do not interfere with their ability to independently perform in daily activities. New touchscreen tools had to be designed for cognitive assessment and had to be at an advanced stage of development but their clinical relevance is still unclear. We aimed to identify digital tools used in the diagnosis of mNCD and assess the diagnostic performance of these tools. **Methods**: In a systematic review, we searched 4 databases for articles (PubMed, Embase, Web of science, IEEE Xplore). From 6516 studies retrieved, we included 50 articles in the review in which a touchscreen tool was used to assess cognitive function in older adults. Study quality was assessed using the QUADAS-II scale. Data from 34 articles were appropriate for meta-analysis and were analyzed using the bivariate random-effects method (STATA software version 19). **Results**: The 50 articles in the review totaled 5974 participants and the 34 in the meta-analysis, 4500 participants. Pooled sensitivity and specificity were 0.81 (95%CI: 0.78 to 0.84) and 0.83 (95%CI: 0.79 to 0.86), respectively. High heterogeneity among the studies led us to examine test performance across key characteristics in a subgroup analysis. Tests that are short and self-administered on a touchscreen tablet perform as well as longer tests administered by an assessor or on a fixed device. **Conclusions**: Cognitive testing with a touchscreen tablet is appropriate for screening for mNCD. Further studies are needed to determine their clinical utility in screening for mNCD in primary care settings and referral to specialized care. This research received no external funding and is registered with PROSPERO under the number CRD42022358725.

## 1. Introduction

Mild neurocognitive disorder (mNCD) is a condition in which people experience cognitive difficulties and dysfunction which do not interfere with their ability to independently perform in daily activities. mNCD may be secondary to neurocognitive diseases like Alzheimer’s disease, Parkinson’s disease, vascular dementia, or others. This condition offers a window of opportunity for cognitive stimulation, treatment of symptoms, implementation of compensatory strategies and introduction of healthier lifestyle habits (diet, exercise, etc.) that may delay the onset of a major neurocognitive disorder [1]. Early diagnosis of these conditions is recommended [2], first and foremost for the personal management of the person and their family, but also to enable rapid management by specialized professionals. However, diagnosing mNCD is challenging because coping mechanisms, types of cognitive deficits, and levels of cognitive reserve vary greatly from one individual to another, resulting in considerable variation in patients’ experiences and symptoms and making it difficult to accurately diagnose this condition [2]. The mNCD and its diagnostic criteria were defined by the DSM-5, and these criteria are very close to those for mild cognitive impairment (MCI), a clinical condition very similar to mNCD, which was widely used before the emergence of mNCD [3]. Early diagnosis can also support clinical research and provide a better understanding of the mechanisms of disease progression or enable participation in clinical trials. The purpose of early diagnosis of mNCD is to slow the progression towards major NCD. Although there is no curative treatment at present, there are numerous strategies that can be implemented to prevent the onset of a major NCD, which is not without consequences for the family and caregivers, with the attendant loss of autonomy for the patient [4,5].

The diagnosis of mNCD relies on medical and neuropsychological evaluation performed in memory centers by a specialized team. Diagnostic criteria have evolved, from mild cognitive impairment (MCI), initially including only memory complaints, to mNCD defined by DSM-5 criteria that now encompasses broader cognitive complaints [3,6]. In both definitions, the person exhibits cognitive deficit identified by cognitive tests and retains autonomy in their daily life. The diagnostic process is long and tedious, often with long waiting times before the first appointment, resulting in a loss of opportunity for the patient. mNCD is insidious, and those affected do not always seek medical attention. The general practitioner (GP) is in the front line when it comes to detecting mNCD and referring to specialized centers [7]. It is therefore important to provide accessible and easy-to-use tools for primary care. Detection is not easy for primary care physicians, since no clearly defined strategy exists to identify people at risk and refer them appropriately to a memory center. The most widely used conventional tests are the Mini Mental State Examination (MMSE) [8] and the Montreal Cognitive Assessment (MoCA) [9]. Both are effective in screening for major neurocognitive disorders [10], but they require training and time and are rarely used by general practitioners. In a systematic review, Chun [11] analyzed the screening tools available for MCI and found that the three most frequently used were the MoCA, the MMSE and the Clock Draw Test (CDT). According to their evaluation criteria, the Six Item Cognitive Impairment Test (6 CIT), the MoCA (with thresholds of ≤24/22/19/15.5) and the MMSE (with a threshold of ≤26) as well as the Hong Kong Brief Cognitive Test (HKBC) were the most effective. However, the authors highlighted the lack of evaluation of these new cognitive tools, with threshold values determined according to the populations and environments in which they are used. The performance of the MMSE and MoCA was compared in the meta-analysis by Pinto [10] and their accuracy in identifying mNCDs was found to be 0.780 (95% CI 0.740–0.820) and 0.883 (95% CI 0.855–0.912), respectively. Both tests have been regularly criticized for their threshold values. Thus, there is still progress to be made in identifying patients with mNCD in primary care [2,10].

Tests are progressively digitized to improve objectivity and speed, with the possibility of automated scoring, which would reduce test-taking time and make them more accessible in primary care for GPs [5,12]. In addition, digital tools make it possible to record more detailed results, such as reaction times or pressure on the screen, which are not accessible to a human assessor. Furthermore, touchscreens are more accessible and intuitive thanks to their direct input, compared to keyboard and mouse use [6,13]. In previous work, we showed good detection of major neurocognitive disorders with touchscreens, which is encouraging for primary care [14].

In the present review and meta-analysis, we aimed to investigate the use of touchscreens for screening for mild cognitive disorders comprising mNCD and MCI, in older adults. We also sought to analyze the performance of these tests in relation to the reference diagnosis.

## 2. Materials and Methods

The protocol was registered with the International Prospective Register of Systematic Review (PROSPERO CRD42022358725), and the report follow the PRISMA-DTA guidelines [15] (see checklist in Appendix A).

### 2.1. Search Strategy

We searched four databases (Medline, Embase, Web of Science and IEEE Xplore) and included all articles published up to 31 December 2024. The last extraction was in April 2025. We used terms relating to screening or diagnosis, older adults, neurocognitive diseases, touchscreen device (see Table A1 in Appendix B). The search terms were broadened to dementia, but we selected only articles dealing with early stages, taking into account the continuum of neurocognitive diseases from MCI/mNCD to dementia. The search strategies were prepared with the help of an experienced librarian. The reference lists of all articles were manually searched to retrieve relevant studies.

### 2.2. Article Selection

We included articles whose participants: (i) were over 60 years of age, (ii) were classified according to the presence of mNCD/MCI determined using a conventional assessment of cognition, based on reference diagnostic criteria (Petersen, National Institute on Aging-Alzheimer’s Association; National Institute of Neurological and Communicative Disorders and Stroke/Alzheimer’s disease and related Disorders Association; Alzheimer’s Disease in neuroimaging initiative, etc.), and (iii) were examined using a novel tool using a digital touchscreen device (tactile tablet, touchscreen computer or smartphone). We did not include studies in which the results for mNCD and M-NCD were mixed and could not be analyzed separately.

### 2.3. Data Extraction

The first two authors independently selected relevant articles from the results of the queries in PubMed, Embase, Web of Science and IEEE Xplore. Any discrepancies were discussed among evaluators until consensus was reached. A third author was consulted in case of disagreement. Reference lists were managed using Zotero^®^ (version 6.0.30) and Excel 2013^®^. Duplicates were individually checked by the two authors. Each investigator evaluated the study selection criteria independently. Reasons for exclusion were noted in Zotero and differences were resolved by discussion.

Descriptive data for each article were collected by two authors and included the descriptive characteristics of the studies, namely: country, year of publication, period of inclusion of participants, mean age of population, reference diagnostic criteria, neuropsychological tests for reference diagnosis. We also recorded the characteristics of the touchscreen test, namely: name of the new test, mode of administration (self-administered or interviewer-administered), cognitive functions assessed, duration of the new test. Sensitivity, specificity and contingency tables were also included for performance analysis in the meta-analysis. If data were missing or unclear to both investigators, they were recorded as “not specified” (NS) in the table. When the contingency table was not included in the original article, we contacted the authors to obtain it, and in the absence of a reply, we calculated the number of true positives, false positives, true negatives and false negatives with sensitivity and specificity from available data.

### 2.4. Quality Assessment

The quality of the included articles was assessed by two authors (NUD, FM) using the Quality Assessment of Diagnostic Accuracy Studies 2 instrument (QUADAS-2) [16] which measures the risk of bias and applicability of diagnostic accuracy studies. It comprises four key domains: patient selection, index test, reference standard, flow and timing. Each domain is considered for its risk of bias and applicability, and judged as high, low or unclear.

There are no official or validated decision rules for determining whether a study is of good or poor quality. We chose to exclude articles that were not of sufficiently high quality, and for this purpose, we defined our own decision rule, namely exclusion of studies with: 2 high risks of bias; or 2 high applicability concerns; or 3 risks of unclear bias; or 2 unclear applicability concerns; or 1 unclear applicability concern and 1 high applicability concern.

### 2.5. Meta-Analysis

We sought to complement the information about the performance of the tools tested. To this end, we collected information on true positives, false positives, true negatives and false negatives. If the information did not appear in an article, we contacted the corresponding author to obtain it.

Meta-analysis was performed with the METADTA program [17] in STATA software (version 19), which uses the bivariate random-effects method. Inter-study heterogeneity was assessed by the I2 coefficient. We performed subgroup analyses according to the type of touchscreen used (touchscreen computer or touchscreen tablet), ease of transport (fixed or mobile device), type of questionnaire administration (rater-administered or self-administered), and test duration (brief test lasting less than 10 min, and longer test lasting more than 10 min).

## 3. Results

### 3.1. Studies Included

The database query yielded 6516 articles. After removal of duplicates and exclusions, 181 articles remained to be evaluated for eligibility. After the QUADAS-2 assessment, we finally included 50 studies in the review and 34 articles in the meta-analysis (Figure 1).

### 3.2. Study Characteristics

We included 50 articles in the systematic review. The characteristics of the included studies are presented in Appendix B. The results are presented in 2 tables according to the digital device used, namely studies using a tactile tablet (Table A2), and studies using a computer touchscreen (Table A3). Articles were published between 2005 [18] and 2024 [19] and were performed in 17 countries located in Europe, Asia, North America and South America.

#### 3.2.1. Participants and Settings

The studies involved 5974 participants (3368 women and 2255 men) (4 studies did not mention the participants’ sex). The number of participants by study varied from 12 [20] to 524 [21] with an average of 119. Mean age of participants was 72 years, and ranged from 53 to 81 years [22,23]. The recruitment was performed in memory centers (*n* = 27), in the community (*n* = 8), in both memory centers and the community (*n* = 3), hospitals (*n* = 14), daycare centers (*n* = 3), health institutions (*n* = 5), memory clinic and research registry (*n* = 2), memory clinic, research registry and community (*n* = 3), hospital, agencies or community advertisements (*n* = 2), hospital, retirement home and community (*n* = 1), nursing home and association (*n* = 1), GP offices and community (*n* = 1), and from a demographic surveillance record (*n* = 1). Three studies did not specify their recruitment methods.

#### 3.2.2. Reference Diagnosis

The reference diagnosis of mNCD/MCI was determined by specialized professionals using reference criteria and tests or parts of tests validated and accepted by the scientific community and are detailed in Appendix B (Table A2 and Table A3). The reference diagnosis was considered as that established by a team of specialists in their own clinic, using official criteria. The studies used diagnostic criteria specific to their usual practice: Petersen’s criteria (*n* = 24), the National Institute on Aging and Alzheimer’s Association (NIA-AA) criteria (*n* = 5), the National Institute of Neurological and Communicative Disorders and Stroke and Alzheimer’s Disease and Related Disorders Association (NINCDS/ADRDA) criteria (*n* = 4), Jak’s criteria (*n* = 1), the National Alzheimer’s Coordinating Center (NACC) criteria (*n* = 1), the American Academy of Neurology (AAN) criteria (*n* = 1), the DSM 5 criteria (*n* = 1), ADI criteria (*n* = 1), NIA-AA and DSM-5 criteria (*n* = 1), Alzheimer’s Disease Neuroimaging Initiative (ADNI) criteria (*n* = 1), Alzheimer’s Disease Research Centers (ADRC) criteria (*n* = 1), and international working group criteria (*n* = 1). Eight studies did not specify diagnostic criteria but reported that a diagnosis was made following a comprehensive medical and neuropsychological evaluation. A sensitivity analysis was carried out to compare the 24 studies using Petersen’s criteria for MCI with the others, and we found no difference between them (see Figure A1 in Appendix B).

#### 3.2.3. Touchscreen Test Procedures

A phase of learning and familiarization with the digital tool was mentioned in 16 studies and was not specified in the others.

Digital test times ranged from 2 min [24] to 2.5 h [25], 13 studies did not specify the duration and one study did not record the test duration [26]. The time needed to complete the tests was less than 5 min in 8 studies, between 10 and 15 min for 7 studies, between 15 and 30 min for 12 studies, between 30 and 60 min for 6 studies and more than an hour in 3 studies.

Thirty-one studies used a self-administered assessment (62%), 13 were assessor-administered (26%) and 6 studies (12%) did not report this information. The professionals involved were health practitioners or researchers trained in the assessments required.

The studies used tactile tablets (*n* = 34) and touchscreen computers (*n* = 16).

Forty of these devices were mobile (80%) versus 6 fixed (12%), while 4 studies did not specify the characteristics of their tool (8%).

#### 3.2.4. Performance Results

Thirty-four studies measured the performance of their digital tests by calculating the sensitivity and specificity of their conclusion compared to the reference diagnosis. Sensitivity ranged from 0.41 (95%CI: 0.21 to 0.64) to 1.00 (95%CI: 0.74 to 1.00) [27,28]. Specificity ranged from 0.56 (95%CI: 0.28 to 0.85) to 1.00 (95%CI: 0.80 to 1.00) [26,28].

### 3.3. Quality Assessment

Overall, the quality of the studies assessed by QUADAS-2 was quite good [18,19,20,21,22,23,24,25,26,27,28,29,30,31,32,33,34,35,36,37,38,39,40,41,42,43,44,45,46,47,48,49,50,51,52,53,54,55,56,57,58,59,60,61,62,63,64,65,66,67,68,69,70,71,72,73,74,75,76,77,78,79] (Table A4 in Appendix B). We excluded 13 studies based on our decision rule.

### 3.4. Meta-Analysis

#### 3.4.1. Main Results

We included 34 articles in the meta-analysis, totaling 4500 participants. Pooled sensitivity and specificity were 0.81 (95%CI: 0.78 to 0.84) and 0.83 (95%CI: 0.79 to 0.86), respectively (Figure 2). The positive likelihood ratio (LR+) was 4.71 (95%CI: 3.88 to 5.73), the negative likelihood ratio (LR-) was 0.23 (95%CI: 0.19 to 0.27), and the diagnostic odds ratio (DOR) was 20.55 (95%CI: 14.66 to 28.80). The summary ROC curve indicated a high overall discriminative performance of the tests, with a summary point near the upper-left corner of the ROC space and reasonably narrow confidence region (Figure 3). I2 coefficient was 56.2, indicating that the studies were quite heterogenous.

#### 3.4.2. Subgroup Analysis

We analyzed the performance of the tests according to their procedures and device characteristics (duration, type of administration, type of touchscreen and mobility of the device) using the chi-2 test. Pooled sensitivity and specificity of these subgroups are presented in Figure 4 and the corresponding forest plots with the individual studies are shown in Appendix B (Figure A2, Figure A3, Figure A4 and Figure A5) and sections below.

##### Duration: Brief Test vs. Longer Test

Sensitivity and specificity in studies using brief tests (0.79; 95%CI: 0.73 to 0.84 and 0.83; 95%CI: 0.76 to 0.88, respectively) were not significantly different from those of studies using longer tests (0.82; 95%CI: 0.77 to 0.86, *p* = 0.68, and 0.84; 95%CI: 0.79 to 0.88, *p* = 0.83) (Figure A2).

##### Type of Administration: Self or Assessor Administered

Sensitivity and specificity in studies using assessor-administered tests (0.81; 95%CI: 0.76 to 0.85 and 0.83; 95%CI: 0.78 to 0.87, respectively) were not significantly different compared to those using self-administered tests (0.81; 95%CI: 0.75 to 0.86, *p* = 0.79, and 0.83; 95%CI: 0.77 to 0.88, *p* = 0.08) (Figure A3).

##### Mobility: Fixed or Mobile Device

Sensitivity in studies using a mobile device were not significantly different from that of studies using a fixed device (0.82; 95%CI: 0.78 to 0.85 and 0.78; 95%CI: 0.66 to 0.86, *p* = 0.43). Conversely, specificity in studies using a mobile device was significantly different higher than in studies using a fixed device (0.85; 95%CI: 0.82 to 0.88 and 0.75; 95%CI: 0.65 to 0.84, *p* = 0.04) (Figure A4).

##### Type of Interface: Touchscreen Computer or Tactile Tablet

Sensitivity and specificity in studies using a tactile tablet (0.81; 95%CI: 0.76 to 0.84 and 0.83; 95%CI: 0.79 to 0.87, *p* = 0.71) were not significantly different from those of studies using a touchscreen computer (0.82; 95%CI: 0.76 to 0.87, and 0.82; 95%CI: 0.76 to 0.87, *p* = 0.68) (Figure A5).

The cognitive tests with the highest combined sensitivity and specificity are summarized in the table below (Table 1).

## 4. Discussion

This review and meta-analysis showed that cognitive tests on touchscreen tools are appropriate to diagnose mNCD in older adults. A large variety of digital devices give satisfactory results in screening for mNCD/MCI. Although imperfect, the overall performance of touchscreen cognitive tests is similar to that of the MoCA, the reference clinical test to screen for mNCD, and several touchscreen cognitive tests outperformed it. However, the heterogeneity of methods and tools makes it difficult to compare studies, precluding any conclusion as to which one is the most effective.

The high degree of heterogeneity among the studies led us to examine test performance based on their main characteristics in a subgroup analysis. It is interesting to note that tests that are short, self-administered and conducted on a touchscreen tablet perform as well as longer tests administered by an assessor or on a fixed device. The former characteristics are very appealing for devices in clinical use, as they are simple, require little professional time and can be used on easily accessible systems.

Through our review, several tools appeared to us to be attractive, due to their good performance in diagnosing mild cognitive disorders (Table 1). Rodríguez-Salgado [54] developed the tool that combines the most practical clinical features and performance, namely the Brain Health Assessment (BHA). It consists of 4 tests: Favorites (associative memory), Match (processing speed and executive function), Line Orientation (visuospatial skills), and Animal Fluency (language). It is a brief, tablet-based cognitive battery validated in English and Spanish, administered by an assessor. Garre-Olmo [28] reported very good results in terms of sensitivity and specificity for the detection of MCI with the Cambridge Cognitive Examination Revised (CAM-COG-R). This is part of a bigger test and consists of 7 tasks assessing cognitive, kinesthetic, visuospatial and motor features on a touchscreen tablet. It can be obtained by purchasing the CAMDEX-DS-II (A Comprehensive Assessment for Dementia in People with Down Syndrome and Others with Intellectual Disabilities) and is available in English and Dutch. The current version is administered by a professional. Park worked on a promising application that revealed the particularities of people with cognitive impairments in their daily use of the telephone keypad [80]. One might imagine downloading this module, which would evaluate keyboard use over several hours or days, taking much of the stress out of traditional exams. Another approach is home assessment, as tested by Thompson with the Mobile Monitoring of Cognitive Change (M2C2) [81], which measures visual working memory, processing speed and episodic memory. The M2C2 is a self-administered test, performed completely remotely, and the episodic memory task demonstrated good ability to distinguish Aß PET status among study participants.

This systematic review and meta-analysis have several limitations. First, it is likely to be affected by publication bias, as studies with null or negative results may be underrepresented. In addition, patient selection in the included studies limits generalizability. Indeed, many of the studies recruited highly selected or convenience samples, which may inflate performance estimates. The predominance of case–control study designs also introduces selection biases that could overestimate diagnostic accuracy compared to prospective cohort study designs. In order to limit potential bias, we excluded 13 articles that we rated, on an ad hoc basis, as having a high risk of bias according to the QUADAS-2 scale, which may also be considered a limitation of our meta-analysis. We also encountered some difficulties with the term “touchscreen device”, which is broad and unclear, as pointed out in Nurgalieva’s review about touchscreen devices. Indeed, devices are not often described in detail, and technology has undergone rapid development in recent years [82]. To address this challenge, we include several terms in our search equation intended to obtain a broad selection of articles and render our screening sensitive (see Table A1 in Appendix B). Nurgalieva’s review also highlights the heterogeneity of older people, and the need to categorize them according to the sensory or cognitive limitations they encounter, in order to be able to propose adapted tools.

## 5. Conclusions

Touchscreen devices can be used to detect mNCD, but their development has yet to be validated by real-life studies. Further efforts are warranted to harmonize assessment methods, although initial results are promising.

In future works, there should be methods for standardizing test procedures so that tools can be compared more easily. It would be of interest for clinical studies to describe their methods accurately and in detail, as well as the manner in which the formal diagnosis was made, in order to fully understand what is being evaluated. Results relating to tool performance are important for the purposes of comparison and should be published in all articles. Touchscreen-based tools need to be evaluated in real-life conditions with people being diagnosed with cognitive disorders, and the results compared.

## Figures and Tables

**Figure 1 diagnostics-15-02383-f001:**
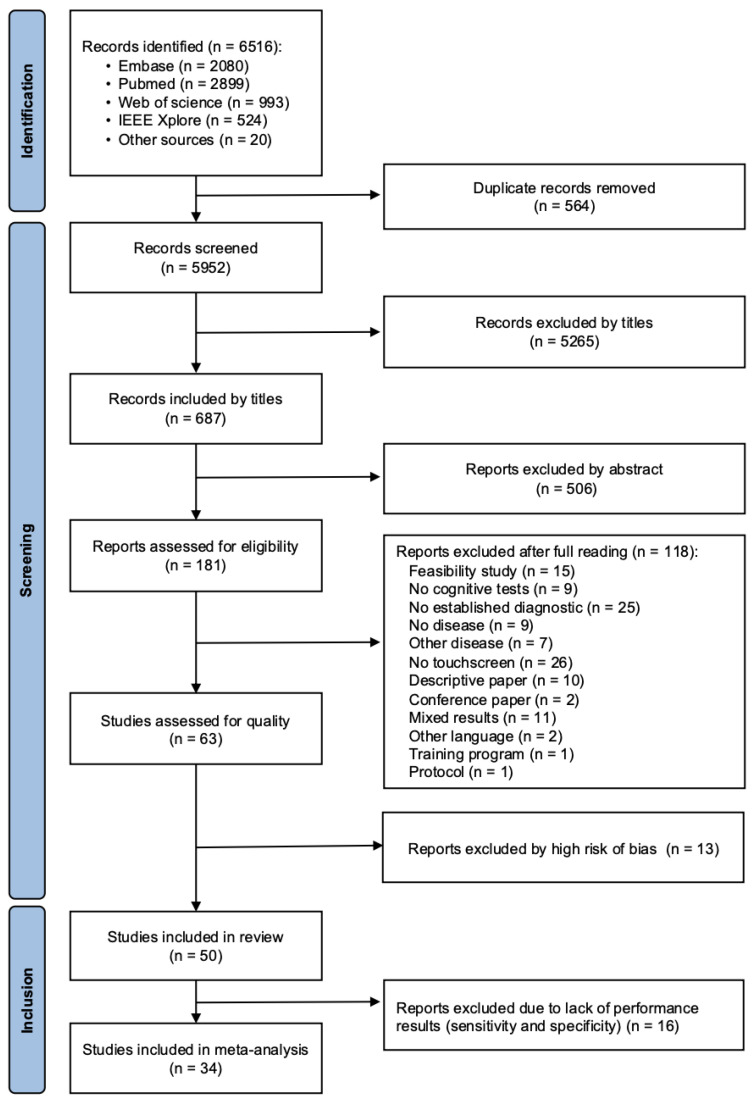
Flow chart of the studies included.

**Figure 2 diagnostics-15-02383-f002:**
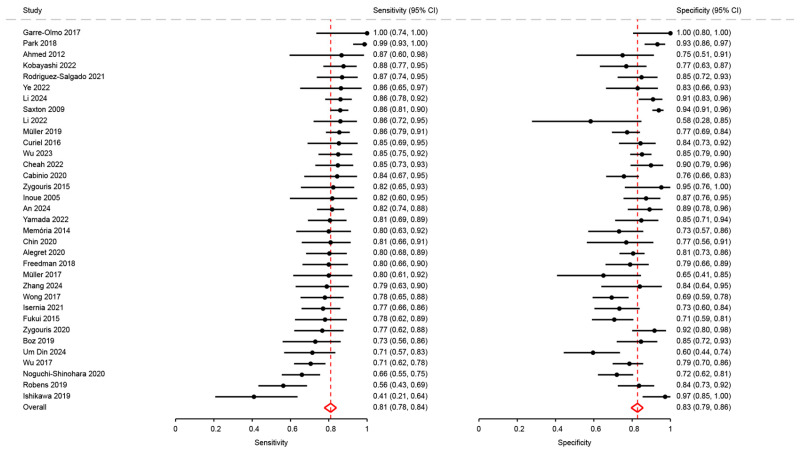
Analysis of sensitivity and specificity for the diagnosis of mild cognitive disorders [18,21,23,26,27,28,29,31,32,34,35,36,37,38,41,43,46,47,48,50,51,53,54,62,63,64,65,66,68,69,72,74,75,76].

**Figure 3 diagnostics-15-02383-f003:**
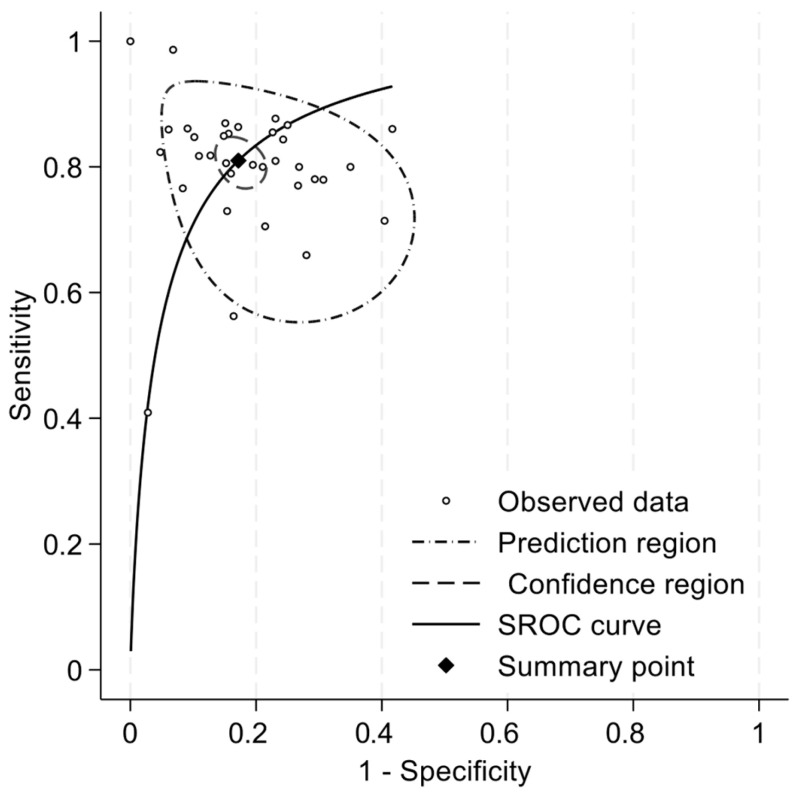
Summary ROC curve of sensitivity and specificity for the diagnosis of mild cognitive disorders.

**Figure 4 diagnostics-15-02383-f004:**
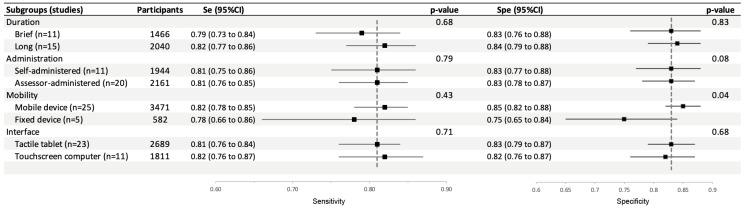
Subgroup analysis of pooled sensitivity and specificity of touchscreen cognitive tests for the diagnosis of mild cognitive disorders.

**Table 1 diagnostics-15-02383-t001:** Cognitive tests with the highest pooled sensitivity and specificity.

Authors Year	Cognitive Testings	Duration (min)	Administration Mode	Cognitive Domains Assessed	Diagnostic Performance
An 2024 [76]	Seoul Digital Cognitive Test	30	NS	Attention, language, visuospatial function, memory, executive function	se: 0.81 spe: 0.89
Cheah 2022 [34]	Rey-Osterrieth Complex Figure	-	Assessor-administered	Visuospatial constructional capabilities and visual memory function (immediate and recall), copying	se: 0.85 spe: 0.91
Curiel 2016 [36]	Miami Test of Semantic Interference and Learning	8–10	NS	Semantic memory, categorization	se: 0.85 spe: 0.84
Garre-Olmo 2017 [28]	7 tasks: figure copying (simple spiral, 3D house, crossed pentagons), clock drawing test, sentence copying, writing a dictated sentence and a spontaneous sentence	10–15	Assessor-administered	Kinesthetic, visuospatial function, motor features	For the task writing a dictated sentence: se: 1.00 spe: 1.00
Li 2024 [74]	Drawing and Dragging Tasks	15	Self-administered	Orientation, selective and sustained attention, visual memory and reconstruction, visuospatial organization, and hand motor skills	se: 0.86 spe: 0.91
Park 2018 [51]	Mobile cognitive function test system for screening mild cognitive impairment	10	Assessor-administered	Memory, orientation, attention, visuospatial ability, language, executive function, reaction time	se: 0.99 spe: 0.93
Rodrigues-Salgado 2021 [54]	Brain Health Assessment	10	Assessor-administered	Memory, processing speed and executive function, visuospatial ability, language	se: 0.87 spe: 0.85
Saxton 2009 [21]	Computer Assessment of Mild Cognitive Impairment	20	Self-administered	Verbal and visual memory, attention, psychomotor speed, language, spatial and executive functioning	se: 0.86 spe: 0.94
Wu 2023 [63]	Efficient Online MCI Screening System	10	Self-administered	Memory, visual attention, flexibility, visuospatial and executive function, cognitive proceeding speed	se: 0.85 spe: 0.85

NS: Not specified.

## Data Availability

The original contributions presented in this study are included in the article/Appendix A. Further inquiries can be directed to the corresponding author.

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
