# Peer review of "Diagnostic Accuracy of Touchscreen-Based Tests for Mild Cognitive Disorders: A Systematic Review and Meta-Analysis"

_diagnostics, 2025, doi:10.3390/diagnostics15182383_

Round 1

Reviewer 1 Report

Comments and Suggestions for Authors

The responses to the following comments should be added to the text at the suitable places of the manuscript.

Comments in Section-2

1.Was the patient selection process free from bias, avoiding inappropriate exclusions or inappropriate case-control designs? Justify your answer.

2.How was the index test conducted and interpreted to avoid bias?

  1. Explain how the reference standard was applied independently of the index test.
  2. How was the index test conducted in a way that reflects its intended use in clinical practice?
  3. Was information on true positives, false positives, true negatives, and false negatives available for all included studies, or were some values obtained by contacting authors? Justify your answer.

6.How was the accuracy of the data obtained from authors verified or handled if data were incomplete or inconsistent?

7.Was the METADTA program in STATA (version 16.1) appropriately used to perform meta-analysis of diagnostic test accuracy, and were all assumptions met? Justify your answer.

8.Were the subgroups (type of touchscreen, ease of transport, administration type, test duration) pre-specified, and were they justified based on clinical or methodological considerations?

  1. How were differences between subgroups assessed?
  2. How were subgroup analyses adequately powered to detect meaningful differences in test performance?

Comments in Section-3

11.Were the reference criteria and tests (e.g., MMSE, MoCA, CERAD) used by specialized professionals validated and consistently applied across all included studies? Justify your answer.

12.How should the pooled sensitivity of 0.81 and specificity of 0.83 be interpreted in terms of clinical relevance for detecting mild neurocognitive disorder (mNCD)?

  1. What factors might explain the moderate heterogeneity observed (I² = 56.4 for sensitivity, 60.7 for specificity) among the included studies?

14.Were the subgroup analyses based on test procedure and device characteristics (duration, administration type, touchscreen type, mobility) pre-specified in the protocol? Justify your answer.

15.Did any subgroups show significantly different pooled sensitivity or specificity compared to others? How were these differences statistically assessed?

16.How might the duration, type of administration, touchscreen type, and mobility of the device impact the diagnostic accuracy in practice?

17.What are the potential limitations related to heterogeneity and study design that could affect the generalizability of the pooled sensitivity and specificity results?

Comments in Section-4

18.How might publication bias have influenced the overall findings of the review, especially regarding the reported performance of touchscreen tools?

19.What was the rationale for excluding studies that did not present an evaluation of their tools, and how might this have affected the comprehensiveness of the review?

20.How did the review address the challenges associated with the broad and unclear definition of the term “touchscreen device”?

21.How does your finding about terminology challenges compare with the observations made in Nurgalieva’s review on touchscreen devices?

22.What suggestions do you have for improving terminology and reporting standards in future research on touchscreen diagnostic tools?

23.How might the identified biases and terminology issues affect the validity and applicability of your review’s conclusions?

General Comments

  1. References 79 and 80 are not cited into the text.

25.Figure captions should be below Figures (for Figures A1, A2, A3, A4)

26.Future work should be the 2nd paragraph of the Conclusion section.
27. Below line 343, the following corrections are to be carried out.

ADRDA- Alzheimer’s Disease and Related Disorders Association

NCD – Neuro Cognitive Disorder

28.In Table A2, the following corrections are to be carried out.
Tablet-based Digital cognitive Test (TDCT)
Digital Clock Drawing Test (DCDT)
Digitized Tree Drawing Test (DTDT)
Efficient Online MCI Screening System (EOMCISS)
Tablet’s Genetic Complex Figure Test (TGCFT)

29.In Table A3, the following correction is to be carried out.
Tablet-based Cancellation Test (TCT)

  1. In References 53 and 64, Journal name is in CAPITAL. Please correct.
  2. In References 59 and 68, year should be bold.

Author Response

  1. Summary

Thank you for taking the time to review our manuscript. Your feedback and comments have helped us to improve our article, which is now more complete. Please find the detailed responses below and the corresponding revisions and corrections highlighted in red in the resubmitted manuscript.

2. Questions for General Evaluation

Reviewer’s Evaluation

Does the introduction provide sufficient background and include all relevant references?

Yes

Is the research design appropriate?

Yes

Are the methods adequately described?

Can be improved

Are the results clearly presented?

Can be improved

Are the conclusions supported by the results?

Can be improved

Are all figures and tables clear and well-presented?

Yes

Comments 1: Was the patient selection process free from bias, avoiding inappropriate exclusions or inappropriate case-control designs? Justify your answer.

Comments 2: How was the index test conducted and interpreted to avoid bias?

Responses 1&2: Thank you for pointing out these important points. We carefully assessed the patient selection process in each of the studies that we reviewed with the QUADAS-2 scale. Our assessment is visible in the first column of the Table A4 of the Appendices. Some of them were excluded due to high risk of bias. Among those retained in the review, we rated 44 of them as being at low risk of bias, 1 as high risk of bias and 5 of unclear risk of bias (2 answers “no”, 31 answers “yes” and 17 answers “unclear” for the item of avoiding inappropriate exclusion, 53 answers “no”, 5 answers “yes” and 5 unclear answers for the item case-control design). The latter studies were kept in our review since they had good assessments on other items of the QUADAS-2 scale. We have mentioned the exclusion of studies with high risk of bias in the flow chart to improve clarity [flow chart, page 5, Figure 1, line 180]. Most of the studies had a case-control design to assess the performances of the digital cognitive test, which also could have introduced bias. The manner in which the index test was performed varied between studies, and due to the case-control design of most studies, it is likely that it was performed after the reference diagnosis. We agree that this is a potential source of bias, mostly related to case/control designs. We have mentioned these potential sources of bias in the Limitations section of the Discussion. [page 10, 4th paragraph, lines 342-344]

Comments 3: Explain how the reference standard was applied independently of the index test.

Response 3: We rated this point in the sub-item of the QUADAS-2 scale and we found that 10 studies were at low risk, 6 at high risk and 47 an unclear risk (as many studies did not report sufficient information on this point).

Comments 4: How was the index test conducted in a way that reflects its intended use in clinical practice?

Response 4: In the studies included, the way in which the index test was performed was probably quite different from its intended use in clinical practice. It is likely that the performance of cognitive tests carried out by clinicians is of lower quality than that carried out by a researcher specializing in neuropsychology. We have addressed this point at the end of the conclusions, emphasizing that evaluating new tests in real life is the best indication of their relevance. [page 11, 2nd paragraph, lines 360-363]

Comments 5: Was information on true positives, false positives, true negatives, and false negatives available for all included studies, or were some values obtained by contacting authors? Justify your answer.

Response 5: Some articles, but not all provided this data. When the contingency table was not directly available in the article, we back-calculated TP, FP, TN, and FN from the sensitivity, specificity and number of subjects. When this calculation was not possible, we contacted the authors (for 15 articles, for which we received 7 answers).

Comments 6: How was the accuracy of the data obtained from authors verified or handled if data were incomplete or inconsistent?

Response 6: When the TP, FP, TN, and FN were available, we computed the calculation of sensitivity and specificity and compared them to that of the paper.

Comments 7: Was the METADTA program in STATA (version 16.1) appropriately used to perform meta-analysis of diagnostic test accuracy, and were all assumptions met? Justify your answer.

Response 7: METADTA is a package for STATA designed to perform meta-analysis of diagnostic studies. It uses the bivariate random-effects method, which is a reference method for this type of meta-analysis. We have added a reference that details this software program (Nyaga 2022) [page 4, 2nd paragraph of the “2.5. Meta-analysis” part, lines 169-170]. The studies are independent, and visual inspection of the forest plot shows that the studies are heterogeneous. On the SROC curve, the points are distributed almost elliptically, which supports the hypothesis of bivariate normality.

We also provide an Excel document that includes data for each study and STATA commands for the analysis in the “Data Availability” statement.

Comments 8: Were the subgroups (type of touchscreen, ease of transport, administration type, test duration) pre-specified, and were they justified based on clinical or methodological considerations?

Response 8: The extraction of information that defined the subgroups was specified in advance and explicitly mentioned in our protocol registered in the Prospero database. This was done in order to describe the studies accurately.

When analyzing the results, we felt it was clinically relevant to compare the performance of cognitive tests in different subgroups in order to detect the potential superiority of certain types of touchscreen tests. In particular, we felt it was important to know whether simpler or shorter tests might be promising for future use in clinical settings.

Comments 9: How were differences between subgroups assessed?

Response 9: Subgroup analyses were performed using chi-square test. We have added a sentence to the manuscript to clarify this point and have clearly added the p-values in Figure 3. [page 8, paragraph “3.4.2. Subgroup analysis”, line 262]

Comments 10: How were subgroup analyses adequately powered to detect meaningful differences in test performance?

Response 10: Thank you for your feedback. In the STATA analysis model, a minimum of four studies is required for the analysis to be relevant, which is the case for our subgroups. We have therefore added the number of studies and the number of participants this represents in Figure 4. [page 8, Figure 4, line 266]

Comments 11: Were the reference criteria and tests (e.g., MMSE, MoCA, CERAD) used by specialized professionals validated and consistently applied across all included studies? Justify your answer.

Response 11: Tables A2 and A3 show that the reference criteria and tests varied considerably from one study to another. We have added a paragraph in the “Results” section to highlight this point. [page 6, paragraph “3.2.2. Reference diagnosis”, line 205 to 219]

Comments 12: How should the pooled sensitivity of 0.81 and specificity of 0.83 be interpreted in terms of clinical relevance for detecting mild neurocognitive disorder (mNCD)?

Response 12: Even if not perfect, the performance of touchscreen cognitive tests seems promising and even better than classical screening tests for clinicians. We have added a sentence to the discussion to alert the reader to these results [pages 9-10, 1st paragraph, and lines 303-305]

Comments 13: What factors might explain the moderate heterogeneity observed (I² = 56.4 for sensitivity, 60.7 for specificity) among the included studies?

Response 13: There is substantial between-study heterogeneity, driven chiefly by the use of very different tests/assays and thresholds across studies, further compounded by differences in populations, settings, and reference standards; this reduces confidence in pooled estimates. As requested by another reviewer (reviewer 2, comment 3), separate I2 coefficients for sensitivity and specificity have been replaced by the generalized I2 coefficient, which is more appropriate. [page 7, section “3.4.1. Main results”, line 253]

Comments 14: Were the subgroup analyses based on test procedure and device characteristics (duration, administration type, touchscreen type, mobility) pre-specified in the protocol? Justify your answer.

Response 14: As mentioned in answer 8, the subgroup analyses were noted as data to be collected for analysis, and we found it relevant to study them in greater depth.

Comments 15: Did any subgroups show significantly different pooled sensitivity or specificity compared to others? How were these differences statistically assessed?

Response 15: The only significant value concerns the specificity of the subgroup on mobile devices.

Univariate comparisons of pooled sensitivity and specificity between subgroups were done using chi-2 test, we have added this information in the results [page 8, section “3.4.2.Subgroup analysis”, line 263]

Comments 16: How might the duration, type of administration, touchscreen type, and mobility of the device impact the diagnostic accuracy in practice?

Response 16: We did not observe any significant differences in performance between the subgroups, with the exception of mobile devices. This encourages the use of simpler and shorter tools for clinical practice.

We chose these characteristics of touchscreen tests to help clinicians who would like to use a digital tool in their daily practice and who would like to use the one with the best performance (according to our hypothesis, a short-self-administered test on a mobile touchscreen). To underline our point, we have included a sentence in the discussion section. [page 10, 2nd paragraph, lines 309 to 313]

Comments 17: What are the potential limitations related to heterogeneity and study design that could affect the generalizability of the pooled sensitivity and specificity results?

Response 17: We are aware that this high degree of heterogeneity limits the generalizability of the pooled sensitivity and specificity results. Indeed, some cognitive tools appear to be very interesting, while others are much more limited. This is partly why we chose to present the studies in descending order of sensitivity in the figures. We would like to point out that this is the evidence available to date, which may be improved upon in future work. We appreciate the Reviewer's comment and have added a sentence to emphasize this point in the discussion. [page 10, 2nd paragraph, line 308-3309 and 4th paragraph, line 337 to 343]

Comments 18: How might publication bias have influenced the overall findings of the review, especially regarding the reported performance of touchscreen tools?

Response 18: The risk of publication bias is inherent in all systematic reviews and meta-analyses, as negative studies are less frequently published. Thus, the overall combined values of sensitivity and specificity may be overestimated due to this bias. However, our clinical research question focused on identifying the most promising tests, which is not affected by such bias. [page 10, 4th paragraph, lines 338 to 343]

Comments 19: What was the rationale for excluding studies that did not present an evaluation of their tools, and how might this have affected the comprehensiveness of the review?

Response 19: When analyzing the full articles selected by our process, we observed that several articles presented a digital cognitive test but did not provide a scientific evaluation of their diagnostic value. They were therefore excluded from the systematic review and meta-analysis. In our view, this does not affect the comprehensiveness of our review as we focus on identifying the test with the most promising diagnostic capacities.

Comments 20: How did the review address the challenges associated with the broad and unclear definition of the term “touchscreen device”?

Responses 20: We agree that exact definition of the term “touchscreen device” may be unclear, which complicates the issue. To address this challenge, we include in our research equation several terms to obtain a broad selection of articles, making our screening sensitive. We are confident about the manual selection of the articles among this wide literature screening. We have added a sentence in the discussion to address this. [page 10, 4th paragraph, lines 349 to 351]

Comments 21: How does your finding about terminology challenges compare with the observations made in Nurgalieva’s review on touchscreen devices?

Response 21: Nurgalieva produced a review based on guidelines for the use of touchscreens in the older adults. The work encountered a number of limits, not least the lack of detail in the descriptions of the touchscreens used. Nurgalieva also points out that, since technology advances rapidly, the terminology is becoming increasingly complex, and validation of the tools was encouraged. This is why we draw a comparison between our difficulty in identifying touchscreens and Nurgalieva's review. We have rephrased the sentence in the discussion to clarify our point. [page 10, 4th paragraph, lines 347 to 349]

Comments 22: What suggestions do you have for improving terminology and reporting standards in future research on touchscreen diagnostic tools?

Response 22: It would be interesting to create a new MESH term to homogenize classification. For clinical studies, it will be important to describe in detail the devices used, as technology is advancing rapidly, and tools no longer look the same. It would be important to include photos of the device in use, or in operation, to fully understand how it works.

Comments 23: How might the identified biases and terminology issues affect the validity and applicability of your review’s conclusions?

Response 23: As mentioned in comment 20, we include in our research equation several terms to obtain a broad selection of articles, making our screening process sensitive. We therefore hope that the biases and terminology issues would have but little effect on the validity. [page 10, 4th paragraph, lines 349 to 351]

Comments 24: References 79 and 80 are not cited into the text.

Response 24: Thank you for pointing out this oversight. We now cite them in the text, after renumbering (reference 79 is now 25 and reference 80 is 27). [page 7, section “3.3. Quality assessment”, line 244]

Comments 25: Figure captions should be below Figures (for Figures A1, A2, A3, A4)

Response 25: We modified the figure captions as required (Figures A1, A2, A3, A4) [pages 22 to 23, Figures A1 to A4, lines 413, 415, 418 and 420]

Comments 26: Future work should be the 2nd paragraph of the Conclusion section.

Response 26: We have moved the paragraph to the conclusion section. [page 11, 2nd paragraph, lines 358 to 364]

Comments 27: Below line 343, the following corrections are to be carried out.

ADRDA- Alzheimer’s Disease and Related Disorders Association

NCD – Neuro Cognitive Disorder

Response 27: We have made these changes as requested. [page 11, section “Abbreviations”, line 391]

Comments 28: In Table A2, the following corrections are to be carried out.

Tablet-based Digital cognitive Test (TDCT)

Digital Clock Drawing Test (DCDT)

Digitized Tree Drawing Test (DTDT)

Efficient Online MCI Screening System (EOMCISS)

Tablet’s Genetic Complex Figure Test (TGCFT)

Response 28: Thank you for these corrections, which have been integrated. We have carefully checked the test names and corrected the errors. To improve readability, we have removed the abbreviations of the index tests, since they appear only once in the table. We have added the language of some index tests which were omitted in the first version of our manuscript. [pages 13 to 19, Tables A2 and A3]

Comments 29: In Table A3, the following correction is to be carried out.

Tablet-based Cancellation Test (TCT)

Response 29: Thank you for your comment. As mentioned above, we have removed the abbreviations of the index tests, since they appear only once in the table. [pages 17 to 19, Table A3]

Comments 30: In References 53 and 64, Journal name is in CAPITAL. Please correct.

Comments 31: In References 59 and 68, year should be bold.

Responses 30 & 31: Thank you, this has been corrected. Note that after renumbering, reference 53 is now 48 and 64 is 57 [page 27, line 555 and line 581], reference 59 is now 53 and 68 is 61 [page 27, line 571 and 592].

  1. Response to Comments on the Quality of English Language

Point 1: The English is fine and does not require any improvement.

Response 1: We appreciate your feedback.

  1. Additional clarifications

-

Reviewer 2 Report

Comments and Suggestions for Authors

The topic of diagnostic accuracy of touchscreen cognitive tests for mild neurocognitive disorder and mild cognitive impairment is clinically relevant. However, there are several issues that need attention before the work can be considered for publication.

1.The manuscript treats mNCD and MCI as interchangeable without a clear specification of the diagnostic reference standards used in the studies (DSM-5, Petersen, NIA-AA, ICD). This introduces heterogeneity that must be explicitly explained and analyzed.

2.Excluding studies based on a self-created QUADAS-2 rule is not methodologically justified. QUADAS-2 is a descriptive tool; exclusion can bias results.

3.The use of separate pooled sensitivity/specificity with I² is not appropriate for diagnostic test accuracy reviews. The recommended methods are bivariate random-effects or HSROC models, which provide a summary point, confidence, and prediction regions, and allow formal investigation of threshold effects.

4.The manuscript follows PRISMA 2020, but DTA reviews require PRISMA-DTA (and PRISMA-DTA for abstracts). STARD 2015 should also be used to assess the completeness of included studies.

5.Currently, there is no SROC plot with confidence and prediction regions, nor are likelihood ratios or diagnostic odds ratios reported. Tables do not list index tests, thresholds, and reference standards in sufficient detail.

6.Terminology: “Sensibility” should be “Sensitivity”; “administrated” should be “administered.”

7.The claim of “no side effects” is misleading; feasibility and usability would be the appropriate outcomes

Author Response

  1. Summary

We thank the Reviewer for the useful suggestions to help us to improve our article. Please find the detailed responses below and the corresponding revisions and corrections highlighted in red in the resubmitted manuscript.

2. Questions for General Evaluation

Reviewer’s Evaluation

Does the introduction provide sufficient background and include all relevant references?

Must be improved

Is the research design appropriate?

Must be improved

Are the methods adequately described?

Must be improved

Are the results clearly presented?

Can be improved

Are the conclusions supported by the results?

Can be improved

Are all figures and tables clear and well-presented?

Can be improved

  1. Point-by-point response to Comments and Suggestions for Authors

Comments 1: The manuscript treats mNCD and MCI as interchangeable without a clear specification of the diagnostic reference standards used in the studies (DSM-5, Petersen, NIA-AA, ICD). This introduces heterogeneity that must be explicitly explained and analyzed.

Response 1: In our review, we used MCI or mNCD as the diagnostic reference, as these concepts are very similar and mNCD represents an evolution of MCI as described by Petersen. Although mNCD is the term that should currently be used, many important studies, particularly older ones, used the MCI classification of cognitive disorders.

We have added a clarification of this point in the introduction. [page 2, 4th paragraph, lines 56 to 59]. In the article selection and results, we have specified the criteria used for the reference diagnosis [page 3, section “2.2. Article selection”, lines 124 to 128 and page 6, section “3.2.2. Reference diagnosis”, lines 207 to 221]. In the tables describing the studies included in the review, this point was clear in the first version of our manuscript.

Nevertheless, we agree with the Reviewer that this could be a potential source of heterogeneity. Therefore, we analyzed this and we now present a sensitivity analysis in the supplementary material, comparing the 24 studies that used Petersen's MCI criteria with the other studies. [page 6, section “3.2.2. Reference diagnosis”, lines 219 to 221 and page 20, Table A4]

Comments 2: Excluding studies based on a self-created QUADAS-2 rule is not methodologically justified. QUADAS-2 is a descriptive tool; exclusion can bias results.

Response 2: We agree with the Reviewer's comment and we are aware that we may have introduced potential for bias here. However, we believe it was controlled, given our objective of analyzing promising tools that were already at an advanced stage of evaluation.

We have added a sentence in the discussion to clarify. [page 10, 4th paragraph, line 345 to 347]

Comments 3: The use of separate pooled sensitivity/specificity with I² is not appropriate for diagnostic test accuracy reviews. The recommended methods are bivariate random-effects or HSROC models, which provide a summary point, confidence, and prediction regions, and allow formal investigation of threshold effects.

Response 3: We agree on this methodological point. In the first version of the manuscript, the meta-analysis was conducted using the bivariate random-effects method, but this was not clearly indicated in Methods. We have now clearly specified this point [page 4, section “2.5. Meta-analysis”, line 171-172]. The separate I2 coefficients for sensitivity and specificity have been replaced by the generalized I2 coefficient, which is more appropriate. [page 7, section “3.4.1. Main results”, line 255]

Comments 4: The manuscript follows PRISMA 2020, but DTA reviews require PRISMA-DTA (and PRISMA-DTA for abstracts). STARD 2015 should also be used to assess the completeness of included studies.

Response 4: Thank you for pointing this out. We have updated the documents with the PRISMA-DTA checklist and it is now available in the Appendices.

Comments 5: Currently, there is no SROC plot with confidence and prediction regions, nor are likelihood ratios or diagnostic odds ratios reported.

Tables do not list index tests, thresholds, and reference standards in sufficient detail.

Response 5: We apologize for this omission. The SROC has now been added [page 8, Figure 3, and line 259] and the likelihood ratios or diagnostic odds ratios are reported [page 7, 3rd paragraph, and lines 251 to 255]

In the Tables, we have already described the index tests in detail, notably by specifying the cognitive dimensions tested. We have also described the criteria and tests used for the reference diagnosis. We did not extract the thresholds from the index test because comparing them would not provide much information. The index tests are different in each study (with the exception of Inbrain Cognitive Screening Test, the Five drawing tasks, the Computer-Administered Neuropsychological Screen for Mild Cognitive Impairment and the Smart Aging Serious Game which appear in 2 studies; Virtual Supermarket Test which appears in 3 studies). A more detailed analysis of the thresholds would be very interesting if a large number of studies used the same index test.

Comments 6: Terminology: “Sensibility” should be “Sensitivity”; “administrated” should be “administered.”

Response 6: We apologize for these mistakes, “sensibility” [page 6, line 219] and “administrated” have been corrected [page 9, 1st paragraph, lines 281 and 283]

Comments 7: The claim of “no side effects” is misleading; feasibility and usability would be the appropriate outcomes

Response 7: The Reviewer is correct, we have removed the offending sentence.

  1. Response to Comments on the Quality of English Language

Point 1: The English could be improved to more clearly express the research.

Response 1: Thank you for your feedback. The article has been thoroughly revised by a native English-speaking medical writer. We are confident that the quality of the English is now much improved.

  1. Additional clarifications

-

Reviewer 3 Report

Comments and Suggestions for Authors

I would like to thank the authors for the effort they have put into this comprehensive and timely review which tackles an important and timely question: how well touchscreen-based tests can detect mild cognitive impairment.

The manuscript is clearly structured, with well-defined aims, solid methodology, and appropriate statistical analysis. I also appreciate the inclusion of subgroup analyses (e.g., test duration, mode of administration, and device type), which add value to the study.

I have some suggestions where the manuscript could be strengthened, in my opinion:

Comparisons: For the subgroup analyses many results are reported simply as “no significant difference.” Even though there are no significant differences, it would be helpful if the authors could comment briefly on what these subgroup findings might mean in practice (for example, whether brief tests might be preferable in primary care if their performance is similar to longer ones). Even short interpretive notes would make the comparisons more meaningful for readers.

Specific tools: The manuscript would benefit from a comparative synthesis of the tools described. A summary table contrasting the most validated tools (e.g., BHA, CAM-COG-R, dCDT, Virtual Supermarket) by test duration, administration mode, cognitive domains assessed, and reported diagnostic performance would make the findings more accessible and useful for readers.

Comments on the Quality of English Language

While the manuscript is understandable, several sentences could be made more concise and grammatically polished to improve readability. Here are a few examples.

  • “A phase of learning and familiarization with the digital tool has been mentioned in 16 studies and was not specified in the other.” (line 194)
  • “We deliberately excluded from our review studies that did not present an evaluation of their tools, which removed a number of interesting touchscreen device that have not yet progressed beyond the feasibility stage.” (lines 296 -297)

Author Response

  1. Summary

We thank the Reviewer for the encouraging feedback and the useful suggestions for improvement. We hope to have responded adequately to all comments. Please find the detailed responses below and the corresponding revisions and corrections highlighted in red in the resubmitted manuscript.

2. Questions for General Evaluation

Reviewer’s Evaluation

Does the introduction provide sufficient background and include all relevant references?

Yes

Is the research design appropriate?

Yes

Are the methods adequately described?

Yes

Are the results clearly presented?

Can be improved

Are the conclusions supported by the results?

Yes

Are all figures and tables clear and well-presented?

Yes

  1. Point-by-point response to Comments and Suggestions for Authors

Comments 1: I would like to thank the authors for the effort they have put into this comprehensive and timely review which tackles an important and timely question: how well touchscreen-based tests can detect mild cognitive impairment.

The manuscript is clearly structured, with well-defined aims, solid methodology, and appropriate statistical analysis. I also appreciate the inclusion of subgroup analyses (e.g., test duration, mode of administration, and device type), which add value to the study.

Response 1: We thank the Reviewer for these positive comments.

Comments 2: I have some suggestions where the manuscript could be strengthened, in my opinion:

Comparisons: For the subgroup analyses many results are reported simply as “no significant difference.” Even though there are no significant differences, it would be helpful if the authors could comment briefly on what these subgroup findings might mean in practice (for example, whether brief tests might be preferable in primary care if their performance is similar to longer ones). Even short interpretive notes would make the comparisons more meaningful for readers.

Response 2: Thank you for this useful suggestion. We added a paragraph of interpretation of the subgroup analyses in the discussion, to highlight our findings. [page 10, 2nd paragraph, lines 310 to 315]

Comments 3: Specific tools: The manuscript would benefit from a comparative synthesis of the tools described. A summary table contrasting the most validated tools (e.g., BHA, CAM-COG-R, dCDT, Virtual Supermarket) by test duration, administration mode, cognitive domains assessed, and reported diagnostic performance would make the findings more accessible and useful for readers.

Response 3: We thank the Reviewer for this idea, and we have now added a new table accordingly. [page 9, Table 1, line 301]

  1. Response to Comments on the Quality of English Language

Point 1: While the manuscript is understandable, several sentences could be made more concise and grammatically polished to improve readability. Here are a few examples.

“A phase of learning and familiarization with the digital tool has been mentioned in 16 studies and was not specified in the other.” (line 194)

“We deliberately excluded from our review studies that did not present an evaluation of their tools, which removed a number of interesting touchscreen device that have not yet progressed beyond the feasibility stage.” (lines 296 -297)

Response 1: We have changed the first example [page 6, line 224-225] and the second has been removed in response to comments from other Reviewers.

In addition, the article has been thoroughly revised by a native English-speaking medical writer. We are confident that the quality of the English is now much improved.

  1. Additional clarifications

-

Round 2

Reviewer 1 Report

Comments and Suggestions for Authors

Responses to all the comments have been addressed. 

Reviewer 2 Report

Comments and Suggestions for Authors

Thank you for addressing all my comments adequately. The manuscript has improved, and I have no further comments.